# New Diterpenes from *Arenga pinnata* (Wurmb.) Merr. Fruits

**DOI:** 10.3390/molecules24010087

**Published:** 2018-12-27

**Authors:** Ji-Fei Liu, Jin-Hai Huo, Chang Wang, Feng-Jin Li, Wei-Ming Wang, Lu-Qi Huang

**Affiliations:** 1Institute of Chinese Materia Medica, Heilongjiang Academy of Chinese Medicine Sciences, Harbin 150036, China; liujifei2019@163.com (J.-F.L.); jinhaihuo@126.com (J.-H.H.); wangchang89993762@126.com (C.W.); wklifengjin@163.com (F.-J.L.); 2National Resource Center for Chinese Materia Medica, China Academy of Chinese Medical Sciences, Beijing 100700, China

**Keywords:** *Arenga pinnata* (Wurmb.) Merr., *ent*-kauran-type diterpene, arenterpenoids

## Abstract

Three new *ent*-kauran-type diterpenes (**1**–**3**), named arenterpenoids A–C, and five known ones (**4**–**8**) were isolated and identified from *Arenga pinnata* (Wurmb.) Merr. Fruits. The structures of these compounds were established by 1D and 2D NMR spectra and HR-ESI-MS. To the best of our knowledge, this is the first scientific report of diterpenes from *Arenga* genus.

## 1. Introduction

*Arenga pinnata* (Wurmb.) Merr. are tall evergreen trees belonging to the genus *Arenga* of the family Palmae. They are widely distributed in Southeast Asian countries, including China. *A. pinnata* fruits are the fruits of the *A. pinnata*. [1]. As a kind of folk medicine, it was first recorded in the Song Dynasty’s “Kai Bao Ben Cao” and the Ming Dynasty’s “Ben Cao Hui Yan” [2,3]. In the folk literature, *A. pinnata* fruits are made into medicinal liquor, which is rapid and significant in relieving pain [4]. *A. pinnata* fruits have significant effects on local neuropathic pain, rheumatism, bone pain and traumatic pain [5]. It is abundant and there is a huge development space [6]. At present, the secondary metabolites from *A. pinnata* fruits have not been reported, and its main effective medicinal ingredients are still not clear. Palm plants contain terpenes, alcohols, alkanes, esters, phenols, quinones, aldehydes and alkaloids, etc [7,8,9]. In recent years, studies have shown that diterpenes have excellent anti-tumor effects in vivo and in vitro [10,11]. The total diterpene of *Rabdosia excisa* has significant inhibitory effects on P-388, H-22, and Lewis B-16 tumor cells [12]; the diterpenes of *Pteris semipinnata* L. have significant inhibitory effects on A 549 and CNE-2 tumor cells [13]. This study is the first separation of the *Arenga* genus by silica gel and ODS column chromatography. We analyzed the chemical constituents of *A. pinnata* fruits and identified three new diterpenes named arenterpenoids A (**1**), B (**2**), and C (**3**) together with known pseudaminic acids (**4**) [14], 12α-(β-d-glucopyranosyl)-7β-hydroxy-kaurenolide (**5**) [15], paniculoside (**6**) [16], agittarioside b (**7**) [17], and orychoside B (**8**) [18]. This report covers the separation and structural analysis of these compounds **1**–**8**
Figure 1.

## 2. Results

Compound **1** possessed the molecular formula of C_26_H_40_O_10_ according to the HR-ESI-MS at *m*/*z* 513.2651 [M + H]^+^. The acid hydrolysis of **1** liberated d-glucose, which was identified by HPLC analysis using an optical rotation detector. ^1^H-NMR spectrum (Table 1) of **1** showed two methyl signals at δ_H_ 1.28 (3H, s), δ_H_ 0.86 (3H, s); two olefinic proton signals at δ_H_ 5.0 (1H, br. s), δ_H_ 5.39 (1H, br. s); an anomeric proton signal at δ_H_ 4.40 (1H, d, *J* = 7.8 Hz). The ^13^C-NMR and DEPT spectrum (Table 1) of **1** showed an *ent*-kauran-type diterpene skeleton [19]. The compound structure has 26 carbon signals, five methylene signals at δc 18.4, 19.8, 29.2, 38.4, 39.2; as well as four quaternary carbon signals at δc 35.2, 43.0, 44.6, 85.4; a carbonyl carbon signal at δc 184.8; and two olefinic carbon signals at δ_C_ 109.7 and 158.0. The sugar moiety consisted of six carbon signals at δc 100.1, 75.2, 78.1, 71.7, 78.0, and 62.9 [20]. The ^1^H-^1^H COSY and HSQC analysis of **1** showed the structures A, B and C (Figure 2). The connectivity of these partial structures and their functional groups were investigated by analysis of HMBC. As shown in Figure 2, the long range correlations were observed between the following protons and carbon signals: H-18 (CH_3_) and C-3, C-4, C-5, C-19 (COOH); H-20 (CH_3_) and C-1, C-5, C-9, C-10; H-15 and C-7, C-8, C-9, C-16; H-14 and C-12, C-13, C-16; H-17 and C-16. Thus, structure **1** was confirmed as shown in Figure 2.

The coupling constants of H-6 (dd, *J*_5,6_ = 6.5 Hz and *J*_6,7_ = 6.5 Hz) and H-7 (d, *J*_6,7_ = 6.5 Hz) observed in the ^1^H-NMR spectrum were the same as those of H-6 (dd, *J*_5,6_ = 6.5 Hz and *J*_6,7_ = 6.5 Hz) and H-7 (d, *J*_6,7_ = 6.5 Hz) in 7β, 16α, 17-trihydroxy-*ent*-kauran-6α, 19-olide [21]. Thus, it was determined that H-6 was in the β-orientation and that H-7 was in the α-orientation. The relative stereochemistry of **1** was assigned by analysis of the NOESY spectrum. The NOESY correlations (Figure 3) of H-6/H-18, H-7/H-20 suggested that the configurations of C-18, C-20 were restricted as 18β, 20α, respectively. Likewise, the NOESY cross-peaks of H-19/H-5, H-5/H-9 and H-9/H-15 showed that the configurations of C-5, C-9 and C-15 were restricted as 5β, 9β and 15β, respectively. Thus, the structure of **1** was determined to be as shown (Figure 1) and elucidated as 7β, 13β-dihydroxy-6α-*O*-β-d-glucopyranosyl-*ent*-kauran-16-en-19-oic acid, named as arenterpenoid A (**1**).

Compound **2** possessed the molecular formula of C_20_H_32_O_7_ according to the HR-ESI-MS at *m*/*z* 385.2186 [M + H]^+^. ^1^H- and ^13^C-NMR spectra indicated that the structure of **2** was similar to that of **1**, except for a 11-methylene group (δ_H_ 1.52 (m); δc 19.8 in **1**), 16-olefinic proton group (δc 158.0 in **1**) and a 17-olefinic proton group (δ_H_ 5.39 (br. s), 5.00 (br. s); δc 109.7 in **1**), as well as presence of a 6β-hydroxy group (δ_H_ 4.54 (dd); δc 85.2 in **2**), a 11β-hydroxy group (δ_H_ 3.95 (m); δc 65.5 in **2**), a 16-homomethyl group (δ_H_ 2.03 (m), δc 43.1 in **2**) and a 17-hydroxy group (δ_H_ 3.75 (d); δc 68.1 in **2**). ^13^C-NMR and DEPT spectrum (Table 1) of 2 showed an *ent*-kauran-type diterpene skeleton [22]. The connectivity of these partial structures and the functional groups were investigated by analysis of HMBC of **2**. As shown in Figure 2, long range correlations were observed between the following protons and carbon signals: H-18 (CH_3_) and C-3, C-4, C-5, C-19 (COOH); H-20 (CH_3_) and C-1, C-5, C-9, C-10; H-14 and C-7, C-8, C-9, C-13; H-15 and C-8, C-15; H-17 and C-13, C-15. Thus, structure **2** was confirmed as shown in Figure 2.

The relative stereochemistry of **2** was assigned by analysis of the NOESY spectrum. The ^1^H- and ^13^C-NMR spectra showed similar data for **1** and **2**. Thus, it was determined that H-6 was in the β-orientation and that H-7 was in the α-orientation. The correlations of H-6/H-18 showed that they were cofacial and were arbitrarily assigned to be β-oriented. H-5, H-9 and H-15 were determined by their correlations with H-18. H-20 and H-11 were determined by their correlations with H-7. Thus, the structure of **2** was determined to be as shown (Figure 1) and elucidated as, 6α, 7β, 11β, 13β, 17-pentahydroxy-*ent*-kauran-19-oic acid, named as arenterpenoid B (**2**).

Compound **3** possessed the molecular formula of C_26_H_42_O_11_ according to the HR-ESI-MS at *m*/*z* 531.2769 [M + H]^+^. The acid hydrolysis of **2** liberated d-glucose, which was identified by HPLC analysis using an optical rotation detector. ^1^H- and ^13^C-NMR spectra indicated that the structure of **3** was similar to that of **2**, except for a 11β-hydroxy group (δ_H_ 3.95; δc 65.5 in **2**) and 17-hydroxy group (δ_H_ 3.75; δc 68.1 in **2**), as well as the presence of a 11-methylene group (δ_H_ 1.36; δc 18.3 in **3**), and 17-(β-glucopyranosyl oxy) group (δ_H_ 4.2, 3.46; δc 76.2 in **3**). The connectivity of these partial structures and the functional groups were investigated by analysis of the HMBC of **3**. As shown in Figure 2, long range correlations were observed between the following protons and carbon signals: H-18 (CH_3_) and C-3, C-4, C-5, C-19 (COOH); H-20 (CH_3_) and C-1, C-5, C-9, C-10; H-15 and C-7, C-8, C-9, C-16; H-7 and C-8, C-14, C-15; H-17 and C-13, C-16, C-1′. Thus, the structure of **3** was confirmed as shown in Figure 2.

The relative stereochemistry of **3** was assigned by analysis of the NOESY spectrum. The ^1^H- and ^13^C-NMR spectra showed similar data for **3** and **2**. Likewise, it was determined that H-6 was in the β-orientation and that H-7 was in the α-orientation. The NOESY of **3** showed the cross-peaks between H-6 and H-18, H-5, H-9; H-7 and H-20. The results showed that the stereochemistry of **3** is similar to the stereochemistry of **2**. Therefore, the structure of **3** was determined to be as shown (Figure 1) and elucidated as, 6α, 7β, 13β-trihydroxy-17-*O*-β-d-glucopyranosyl-*ent*-kauran-19-oic acid, named as arenterpenoid C (**3**). The other compounds were characterized as pseudaminic acid (**4**), 12α-(β-d-glucopyranosyl)-7β-hydroxy-kaurenolide (**5**), paniculoside (**6**), agittarioside b (**7**), orychoside B (**8**) by comparing their NMR spectroscopic data with the literature values (Appendix A).

All these compounds are reported here for the first time in *Arenga* genus. The kaurane type diterpene is a kind of tetracyclic diterpene with hydrogenated phenanthrene as the mother nucleus [23]. According to the structural rule of the kauri-type diterpene, the compounds **1**–**8** are all C-20 unoxidized kauri-type. Most of such structures isolated and artificially synthesized in plants have significant biological activities, such as antimicrobial activity and cytotoxicity [24]. This study provides an experimental and scientific basis for drug design and discovery in *A. pinnata* fruits.

## 3. Experimental Section

### 3.1. General Experimental Procedures

NMR spectra were measured on a Bruker AV-400 spectrometer (Bruker Company, Waltham MA, USA) with TMS as an internal standard. High-resolution ESI-MS mass spectra were carried out on an AB SCEIX Triple-TOFTM 5600^+^ instrument (A.B. Company, Milwaukee, WI, USA). UV spectra were recorded on a PerkinElmer Lambda UV-365 instrument (PE Company, Waltham MA, USA). IR spectra were recorded on a PerkinElmer Spectrum Two spectrometer (PE Company, Waltham MA, USA) with KBr disks. Preparative HPLC (515-2414, Waters, Milford, CT, USA) was performed on 5C18 MS-II (10 μm, 20 × 250 mm, cat. no.: 38024-01, COSMOSIL, Tokyo, Japan). NH_2_ column (4.6 × 250 mm, cat. no.: S1119, Kasei Company, Tokyo, Japan), silica gel (200–300 mesh, Haiyang Co, Qingdao, China), Amberlite IRA-400 (OH-, Alfa Aesar, Heysham, UK) and ODS (50 µm, AAG12S50, YMC Company, Kyoto, Japan) were used for column chromatography. Detectors (2424, ELS, Waters) and (2998, PDA, Waters) were used in the HPLC. Precoated silica GF 254 plates (Haiyang Company, Qingdao, China) were used for TLC analysis. All the solvents were of analytical grade (Tianjinfuyu Company Ltd., Tianjin, China).

### 3.2. Plant Material

The *A. pinnata* fruits were collected from Guangxi in China during September 2017, and authenticated by Prof. Weiming Wang of the Heilongjiang Research institute of Chinese Medicine. The fruitage had been deposited at the Heilongjiang Research institute of Chinese Medicine.

### 3.3. Extraction and Isolation

The *A. pinnata* fresh fruits (30.0 kg) were extracted with 70% EtOH (200 L × 3 h × 3 times). The combined extract was concentrated under vacuum yielding a residue (3.0 kg) which was dissolved in H_2_O (12 L) and extracted sequentially with petroleum, chloroform, ethyl acetate and n-butanol (12 L × 3 h × 5 times). The eluate was separately concentrated in vacuo to give a petroleum syrup (109.0 g), chloroform syrup (123.0 g), ethyl acetate syrup (205.0 g), and an n-butanol syrup (380.0 g). In this study, we only separated the n-butanol layer. The n-butanol (380.0 g) extract was subject to column chromatography on silica gel (4460.0 g) and eluted with CH_2_Cl_2_/MeOH (20:1 (80.0 L), 10:1 (110.0 L), 5:1 (120.0 L), 3:1 (100.0 L), 2:1 (80.0 L) and 1:1 (60.0 L), *v*/*v*) to afford six fractions (fractions A (36.0 g), B (96.7 g), C (90.2 g), D (40.1 g), E (21.2 g), F (20.1 g). The TLC and HPLC were used to observe each of the fractions, and similar fractions were combined to afford A1-A6, B1-B5, C1-C6, D1-D6, E1-E10, F1-F6. Fraction B4 (21.7 g) was eluted by Rp-18 (600.0 g) (MeOH/H_2_O 2:8 (1.4 L)→3:7 (2.0 L)→4:6 (2.7 L)→5:5 (2.0 L)→6:4 (2.0 L)→7:3 (1.4 L)→8:2 (1.4 L)→9:1 (0.8 L)→1:0 (1.0 L), *v*/*v*) to afford nine subfractions (subfractions B4-1–B4-9). Subfraction B4–6 further purified by a preparative RP-HPLC (55% MeOH/H_2_O, flow rate 5 mL/min) to give **1** (100.12 mg, t_R_ = 23 min). Subfractions B4–5 were further purified by a preparative RP-HPLC (55% MeOH/H_2_O, flow rate 5 mL/min) to give **2** (9.52 mg, t_R_ = 42 min). Fraction C1 (8.0 g) was eluted by Rp-18 (600.0 g) (MeOH/H2O 2:8 (1.4 L)→3:7 (2.0 L)→4:6 (2.7 L)→5:5 (2.7 L)→6:4 (2.0 L)→7:3 (1.4 L)→8:2 (1.4 L)→9:1 (0.8 L)→1:0 (1.0 L), *v*/*v*) to afford nine subfractions (subfractions C1-1–C1-9). Subfractions C1–3 were further purified by a preparative RP-HPLC (20% MeOH/H_2_O, flow rate 5 mL/min) to give **4** (11.89 mg, t_R_ = 30 min). Subfraction C1–5 were further purified by a preparative RP-HPLC (45% MeOH/H_2_O, flow rate 5 mL/min) to give **5** (8.50 mg, t_R_ = 25 min). Fraction D3 (18.0 g) was eluted by Rp-18 (600.0 g) (MeOH/H_2_O 2:8 (1.4 L)→3:7 (2.0 L)→4:6 (2.7 L)→5:5 (2.7 L)→6:4 (2.7 L)→7:3 (1.4 L)→8:2 (1.4 L)→9:1 (0.8 L)→1:0 (1.0 L), *v*/*v*) to afford nine subfractions (subfractions D3-1–D3-9). Subfractions D3–6 were further purified by a preparative RP-HPLC (40% MeOH/H_2_O, flow rate 5 mL/min) to give **3** (10.61 mg, t_R_ = 32 min). Subfraction D3–6 were further purified by a preparative RP-HPLC (45% MeOH/H_2_O, flow rate 5 mL/min) to give **8** (4.51 mg, t_R_ = 41 min). Subfractions D3–4 were further purified by a preparative RP-HPLC (60% MeOH/H_2_O, flow rate 5 mL/min) to give **6** (70.67 mg, t_R_ = 29 min). Subfractions D3–5 were further purified by a preparative RP-HPLC (60% MeOH/H_2_O, flow rate 5 mL/min) to give **7** (4.32 mg, t_R_ = 37 min).

Arenterpenoids A (**1**). Yellow amorphous powder. Gave ${\left[\alpha \right]}_{D}^{25}$ +10.0 (c = 1.76, MeOH); IR (KBr) 3436, 2946, 2835, 1719, 1635, 1467, 1231, and 1069 cm^−1^; ^1^H- and ^13^C-NMR (MeOH, 400, 100 MHz) data, see Table 1; HR-ESI-MS *m*/*z* 513.2651 [M + H]^+^ (calc. for C_26_H_40_O_10_, 513.2655) (Figure 1 and Figure 2).

Arenterpenoids B (**2**). Yellow amorphous powder. Gave ${\left[\alpha \right]}_{D}^{25}$ −31.6 (c = 0.66, MeOH); IR (KBr) 3381, 2931, 1730, 1664, 1451, 1229, and 1039 cm^−1^; ^1^H- and ^13^C-NMR (MeOH, 400, 100 MHz) data, see Table 1; HR-ESI-MS *m*/*z* 385.2186 [M + H]^+^ (calc. for C_20_H_32_O_7_, 385.2182) (Figure 1 and Figure 2).

Arenterpenoids C (**3**). White amorphous powder. Gave ${\left[\alpha \right]}_{D}^{25}$ −21.6 (c = 0.48, MeOH); IR (KBr) 3358, 2949, 1716, 1662, 1460, 1225, and 1070 cm^−1^; ^1^H- and ^13^C-NMR (MeOH, 400, 100 MHz) data, see Table 1; HR-ESI-MS *m*/*z* 531.2769 [M + H]^+^ (calcd. for C_26_H_42_O_11_, 531.2761) (Figure 1 and Figure 2).

### 3.4. Acid Hydrolysis and HPLC Analysis

The isolated compounds (**1**,**3**) (2.0 mg) were in 1.0 mL HCl and were each heated under reflux for 3 h. After cooling, the two mixtures were separately filtered with Amberlite IRA-400 to give a solution. Assigned with AcOEt to get two layers. The aqueous layer was evaporated to dryness under vacuum, and then subjected to HPLC analysis using an NH_2_ column and an optical-rotation detector. d-glucose was confirmed by comparison of the t_R_ with that of an authentic sample (mobile phase: MeCN/H_2_O 85: 15 (*v*/*v*); flow rate: 0.8 mL/min; t_R_ = 12.8 min (d-glucose, positive optical rotation))

## 4. Conclusions

As a traditional Chinese medicine, *A. pinnata* fruits were mainly used to treat rheumatism and bone pain. This study obtained eight diterpene compounds from *A. pinnata* fruits, including three new diterpenes and five known ones. This also reveals some structural characteristics of the chemical constituents in the *A. pinnata* fruit, which provides some clues for further clarifying the composition of the components and correlations of the relative plant species. In this study we have made this contribution to discover active ingredients and leading compounds and additionally provided an experimental and scientific basis of drug design and drug discovery of the *A. pinnata* fruits.

## Figures and Tables

**Figure 1 molecules-24-00087-f001:**
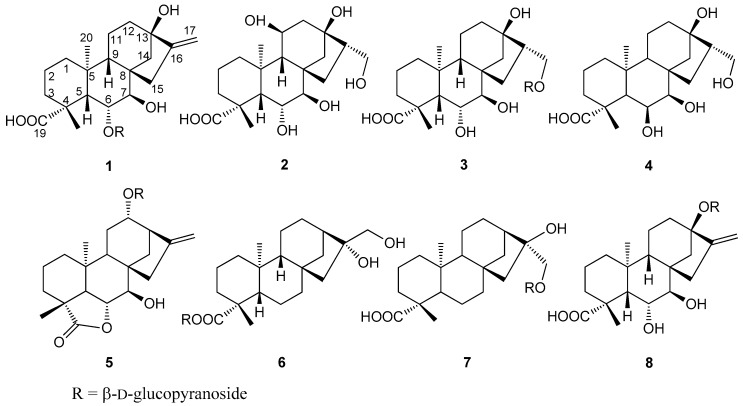
Structures of compounds **1**–**8** from of *Arenga pinnata* (Wurmb.) Merr. ruits.

**Figure 2 molecules-24-00087-f002:**
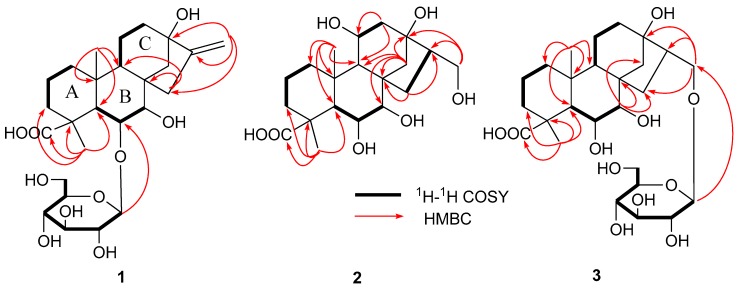
Key HMBC and ^1^H-^1^H COSY correlations of compound **1**–**3**.

**Figure 3 molecules-24-00087-f003:**
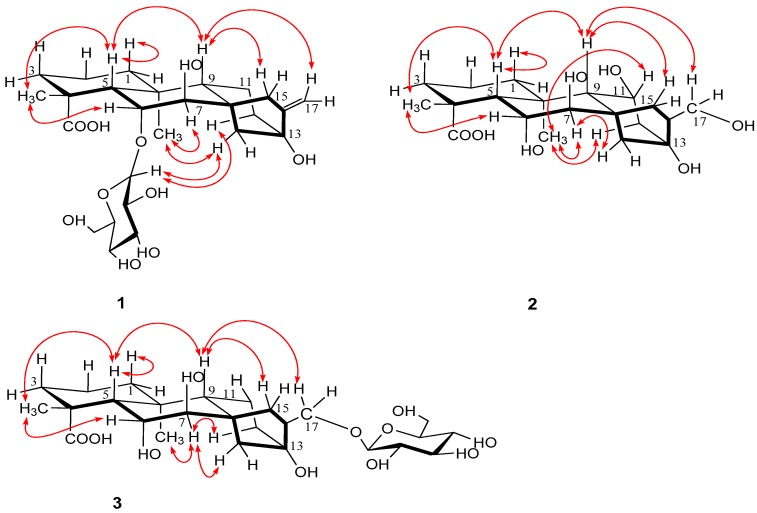
Key NOESY correlations of compound **1**–**3**.

**Table 1 molecules-24-00087-t001:** ^1^H- and ^13^C-NMR Data of **1**–**3** (CD_3_OD).

NO.	1	2	3
δ_H_ (J, Hz)	δ_C_	δ_H_ (J, Hz)	δ_C_	δ_H_ (J, Hz)	δ_C_
1	1.53 (m)	38.4	1.88 (m)	37.1	1.06 (m)	38.4
	1.04 (m)		1.73 (m)		1.55 (m)	
2	1.53 (m)	18.4	1.41 (m)	18.6	1.54 (m)	18.3
3	2.01 (m)	29.2	1.83 (m)	24.1	2.00 (m)	29.2
	1.40 (m)		1.65 (m)		1.42 (m)	
4		43.0		43.2		43.1
5	1.88 (d, 6.5)	52.7	1.85 (d, 7.3)	51.4	1.82 (d, 6.5)	52.8
6	4.62 (dd, 6.5, 6.5)	84.9	4.54 (dd, 7.3, 7.3)	85.2	4.56 (dd, 6.5, 6.5)	85.2
7	4.26 (d, 6.5)	72.2	4.14 (d, 7.3)	72.3	4.23 (d, 6.5)	72.1
8		44.6		48.2		48.0
9	1.17 (dd, 4.9, 12.1)	56.2	1.17 (dd, 4.7, 12.1)	59.7	1.17 (m)	58.8
10		35.2		37.2		35.4
11	1.52 (m)	19.8	3.95 (m)	65.5	1.36 (m)	18.3
12	2.31 (dd, 7.5, 13)	39.2	1.76 (m)	49.7	1.79 (m)	24.4
			1.37 (m)		1.68 (m)	
13		85.4		86.0		85.4
14	2.05 (d, 10.9)	39.9	2.12 (d, 13.0)	47.5	2.17 (d, 13.2)	47.6
	1.74 (m)		1.16 (d, 13.0)		1.16 (d, 13.2)	
15	2.81 (br. d, 14.9)	42.0	2.05 (m)	33.5	2.09 (m)	33.6
	1.78 (m)		1.49 (d, 12.6)		1.53 (m)	
16		158.0	2.03 (m)	43.1	2.05 (m)	43.8
17	5.39 (br. s)	109.7	3.75 (d, 11.5)	68.1	4.27 (d, 10.4)	76.2
	5.00 (br. s)		3.58 (d, 11.5)		3.46 (d, 10.4)	
18	1.28 (s)	25.9	1.34 (s)	24.7	1.27(s)	25.8
19		184.8		185.1		185.0
20	0.86 (s)	20.9	0.98 (s)	23.4	0.85 (s)	20.9
1′	4.40 (d, 7.8)	100.1			4.28 (d, 7.7)	100.1
2′	3.16 (m)	75.2			3.21 (m)	75.2
3′	3.37 (m)	78.1			3.37 (m)	78.1
4′	3.28 (m)	71.7			3.28 (m)	71.7
5′	3.28 (m)	78.0			3.28 (m)	78.0
6′	3.83 (dd, 2.1, 12.0)	62.9			3.88 (dd, 2.1, 12.0)	62.9
	3.59 (m)				3.65 (m)

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
