# Peer review of "New Diterpenes from Arenga pinnata (Wurmb.) Merr. Fruits"

_molecules, 2018, doi:10.3390/molecules24010087_

Round 1

Reviewer 1 Report

This study identifies three new ent-kaurane type diterpenoids from A. pinnata fruits. Use of 2D NMR spectroscopy along with comparisons to known compounds have been used to confirm structure. As the NMR data is not supplied it is difficult to ascertain the accuracy of the assignments but seem plausible enough.  There are a few minor comments and corrections that follow 

It may have been relevant to explain what this fruit is to a reader that is outside SE asia and knows nothing about them 

The extraction process is not entirely clear and a diagram may assist here. What mass of fruit was used in the initial extraction (line125)? how was the extraction carried out? was it just left sitting in the solvent? Lines 127-129 the initial ethanolic extract are redissolved in water and partitioned with various solvents but then the EtOH elution was concentrated. What is this? It is not clear-  is it the leftovers after the partitioning? Or was EtOH also used. What happened to the other partitioned layers- this paragraph needs work  

The syrup (380g, which seems a lot to try column chromatography on) was then eluted to give six fractions- what volume of each eluant was used (I know its not common to ask but if I was to try and repeat this work I would want to have an idea?) Was there any issues using such a high concentration of methanol in the eluant? 

Both Fraction B and D were eluted by Rp-18- I am assuming this is the ODS in the general experimental procedure but had trouble finding exactly what this was (product code etc) from the YMC website- again no mention of volumes 

What guided the authors to each of the fractions and subfractions- was it TLC?- what was used to visualise it. What detector was used in the HPLC? What wavelength? 

Figure 1- compound 3 is missing the OR group of the sugar 

What was the purpose of the acid hydrolysis and HPLC analysis- not mentioned in the substructure elucidation section. Was the HPLC done on the same system as used for PrepHPLC- if not needs to be included. Line 157- ‘Assigned with AcOEt to get two layers. It was evaporated’. Which layer was evaporated? Was it extracted with EtOAc or washed with it 

Line 33 Replace ‘Now, the experiment was’ with This study is … 

Line 34 replace ‘We tried to analyze’ with ‘We analyzed ‘... 

Line 37 an ‘and’ between (7) and (8) 

Line 38 it discusses the separation and structural analysis of compounds 1-8 yet compounds 4-8 are not discussed at all in the experimental or results. Again a labelled diagram showing the extraction process and where each compound found would be useful 

Line115 ‘on a sunfire C18 prep column and analytical HPLC using an amine column (cat number?)

Author Response

I’d like to submit the revised manuscript entitled “New diterpenes from Arenga pinnata (Wurmb.) Merr. fruits” for publication in Molecules.

We thank the reviewers for their valuable comments on previous manuscript. We have carefully taken their comments into consideration in preparing our revision. The changes have been highlighted in red in the revised manuscript.

The reply has been described in detail in the responses to the reviewers. 

We would like to express our great appreciation to you and reviewers for comments on our paper. Looking forward to hearing from you.

Thank you and best regards.

Yours sincerely

Wei-Ming Wang

Reviewer 2 Report

This manuscript reports the isolation and identification of three new diterpenes from A. pinnata, a traditional Chinese medicine. Five known compounds are also isolated and characterised. The spectroscopic identification is largely based on 2D NMR and HRMS and appears to have been done well. The paper is suitably concise but leads the reader carefully through the logical train of evidence and deduction towards the structures.

I have two minor issues that the authors should address before final acceptance.

The Supplementary Information contains copies of the "D NMR spectra but good quality copies of the 1H and 13C 1D spectra should also be presented, to allow the reader to assess the data for themselves and to prove the purity.

2. It is unfortunate that no biological data were available for these diterpenes, in view of the claimed traditional medical uses of the plant. This is clearly a short preliminary paper but some indication of the bioactivity (or lack thereof) would be useful. This is only a suggestion, not a requirement.

Author Response

Dear Editor,

I’d like to submit the revised manuscript entitled “New diterpenes from Arenga pinnata (Wurmb.) Merr. fruits” for publication in Molecules.

We thank the reviewers for their valuable comments on previous manuscript. We have carefully taken their comments into consideration in preparing our revision. The changes have been highlighted in red in the revised manuscript.

The reply has been described in detail in the responses to the reviewers. 

We would like to express our great appreciation to you and reviewers for comments on our paper. Looking forward to hearing from you.

Thank you and best regards.

Yours sincerely

Wei-Ming Wang
